# A Probiotic *Bacillus amyloliquefaciens* D-1 Strain Is Responsible for Zearalenone Detoxifying in Coix Semen

**DOI:** 10.3390/toxins15120674

**Published:** 2023-11-28

**Authors:** Tao Deng, Yefei Chen, Jinqiang Zhang, Yanping Gao, Changgui Yang, Weike Jiang, Xiaohong Ou, Yanhong Wang, Lanping Guo, Tao Zhou, Qing-Song Yuan

**Affiliations:** 1Resource Institute for Chinese & Ethnic Materia Medica, Guizhou University of Traditional Chinese Medicine, Guiyang 550025, China; td18300850597@163.com (T.D.); jqz18215598296@139.com (J.Z.); gaoyanping1121@foxmail.com (Y.G.); cgy15285135045@163.com (C.Y.); jwk_88@163.com (W.J.); yanhong824@hotmail.com (Y.W.); 2National Resource Center for Chinese Meteria Medica, State Key Laboratory for Quality Ensurance and Sustainable Use of Dao-di Herbs, Beijing 100700, China; glp01@126.com

**Keywords:** zearalenone, ZEN-detoxifying strain, *Bacillus amyloliquefaciens*, coix semen

## Abstract

Zearalenone (ZEN) is a mycotoxin produced by *Fusarium* spp., which commonly and severely contaminate food/feed. ZEN severely affects food/feed safety and reduces economic losses owing to its carcinogenicity, genotoxicity, reproductive toxicity, endocrine effects, and immunotoxicity. To explore efficient methods to detoxify ZEN, we identified and characterized an efficient ZEN-detoxifying microbiota from the culturable microbiome of *Pseudostellaria heterophylla* rhizosphere soil, designated *Bacillus amyloliquefaciens* D-1. Its highest ZEN degradation rate reached 96.13% under the optimal condition. And, D-1 can almost completely remove ZEN (90 μg·g^−1^) from coix semen in 24 h. Then, the D-1 strain can detoxify ZEN to ZEM, which is a new structural metabolite, through hydrolyzation and decarboxylation at the ester group in the lactone ring and amino acid esterification at C2 and C4 hydroxy. Notably, ZEM has reduced the impact on viability, and the damage of cell membrane and nucleus DNA and can significantly decrease the cell apoptosis in the HepG2 cell and TM4 cell. In addition, it was found that the D-1 strain has no adverse effect on the HepG2 and TM4 cells. Our findings can provide an efficient microbial resource and a reliable reference strategy for the biological detoxification of ZEN.

## 1. Introduction

Zearalenone (ZEN), chemically designated as 6-[10-hydroxy-6-oxo-trans-1-undecenyl]-B-resorcyclic acid lactone, was first identified from moldy com in 1962 [1,2]. It is a non-steroidal estrogenic mycotoxin produced by the genus *Fusarium*, mainly including *F. graminearum* (*Gibberella zeae*), *F. culmorum*, *F. equiseti*, *F. crookwellense*, and *F. semitectum* [3,4,5]. Under favorable environmental conditions for fungus growth, in pre- or postharvest, cereal crops and grains are easily infected with *Fusarium* sp., especially *F. graminearum*, leading to the accumulation of high levels of ZEN, especially in coix semen, maize, wheat, and barley [6,7,8]. Currently, *Coix lacryma jobi* L. belongs to the family Poaceae. Its coix semen, as a miscellaneous food and traditional Chinese medicine (TCM) in Asia, can be severely polluted with ZEN, over 120 μg·kg^−1^ in severe cases [3].

Oral intake of the ZEN-contaminated cereals and grains is leading to several reproductive health issues due to competitively binding to estrogen receptors ERα and ERβ in humans and animals [4,9]. And, prolonged dietary ZEN exposure is bringing carcinogenicity (liver cancer, testicular cancer, breast cancer, and esophageal cancer), genotoxicity, reproductive toxicity, endocrine effects, and immunotoxicity [10,11]. Furthermore, ZEN was isolated from the endometrial tissue of women suffering from adenocarcinoma and some endometrial hyperplasia cases [1]. The International Agency for Research on Cancer (IARC) listed ZEN and its derivatives as the third-largest category of carcinogens in 2002 [12]. The World Health Organization (WHO) and the World Agricultural Organization (FAO) regulate the maximum tolerable daily intake of ZEN in the human body as 0.5 μg·mL^−1^ [12]. According to China’s Food Hygiene Standard GB2716-2017 [12], the maximum limitations of ZEN in grains, and their products for humans, is 60 μg·kg^−1^.

Efficient methods to detoxify ZEN are desirable to reduce economic losses and resolve issues related to food safety, because widespread ZEN contamination in feed and food can occur in problematic years [1]. Many ZEN-removal methods including physical, chemical, and biological methods have been explored [13]. As physico-chemical methods, including washing, heat treatment, adsorbing, ozone detoxification, hydrogen peroxide treatment, alkali treatment, sodium carbonate, and soaking treatment, can lead to loss of nutrients and secondary hazards, the ZEN cannot be efficiently detoxified; therefore, they are not ideal ways to remove ZEN. So far, these methods are not practical when applied to food or feed detoxification of ZEN [6,14,15]. The biotransformation of mycotoxins to non-toxic metabolites using pure cultures of microorganisms, especially probiotics, or using cell-free enzyme preparation is an attractive possibility [16].

Isolation of effective and safe microorganisms to convert ZEN into non-toxic metabolites is the key to biotransformation methods. As the metabolites α-ZOL and β-ZOL are still estrogenic cytotoxins, the microbial transformation of ZEN to them cannot be regarded as detoxification [17,18]. Also, formation of ZEN-mono-glucosides and -di-glucosides [19,20] and ZEN-sulfate [4,21] cannot be considered real detoxification but rather the formation of mimicked mycotoxins, because the conjugates may be hydrolyzed during digestion in the intestine [22,23], releasing ZEN again [24]. Cell proliferation assay proved that the cleavage product of ZEN in the lactone undecyl ring system did not show any estrogenic activity in the human breast cancer MCF-7 cell and resulted in permanent detoxification [25,26]. It has been shown in previous studies that some microorganisms can degrade and assimilate ZEN. *Gliocladium roseum* was the first strain identified to transform ZEN to a novel nonestrogenic metabolite [27]. In the past decade, it has been found that *Bacillus* spp. (i.e., *B. subtilis*, *B. licheniformis*, *B. velezensis*, and *B. cereus*), *Lactobacillus* spp. (i.e., *L. rhamnosus*, *L. plantarum*, *L. reuteri*, *L. mucosae*, and *L. paracasei*), *Saccharomyces cerevisiae,* and *Trichosporon mycotoxinivorans* [25] also have the ability to detoxify or assimilate ZEN [28]. However, efficient detoxification microorganisms, transforming ZEN to form non-toxic new metabolites, have rarely been reported.

It has been shown in previous studies that *B. amyloliquefaciens* strains possess the detoxification ability of aflatoxins B1 and ochratoxin [29,30]. In addition, *B. amyloliquefaciens* with probiotic potentials is broadly used for infant milk, fermented food, gastrointestinal regulators [31], and domestic-animal feed [32,33] as additives [34,35]. It has been proven to have multiple functions, such as alleviating inflammatory bowel disease, and improving growth performance and immunity. A few *B. amyloliquefaciens* strains reduce fungal toxins through adsorption which has also has been reported, rather than through biotransformation [34]. However, the degradation and detoxification of ZEN using *B. amyloliquefaciens* has not been explored before.

In the present study, an efficient ZEN-detoxifying microbiota, designated D-1, was identified and characterized as *B. amyloliquefaciens* from the culturable microbiome of *Pseudostellaria heterophylla* rhizosphere soil. The D-1 strain can transform ZEN to 1-(3-prolinel,5-isoleucinelphenyl)-10′-hydroxy-1-undecen-6-one or 1-(3-isoleucinel,5-prolinelphenyl)-10′-hydroxy-1-undecen-6-one (named ZEM) through hydrolyzation and decarboxylation at the ester group in the lactone ring, and amino acid esterification at C2 and C4 hydroxy. The main difference between these two metabolites is the esterification positions of isoleucine and proline at the C2 and C4 hydroxyl groups. Notably, ZEM has reduced the impact on viability, and the damage to the cell membrane and nucleus DNA, through decreasing the leakage of LDH, the content of intracellular MDA, and the activity of intracellular SOC, and can significantly decrease the cell apoptosis via inhibiting the activation of the ER stress pathway in the HepG2 cell and TM4 cell. In addition, it was found that the D-1 strain has no adverse effect on the HepG2 and TM4 cells.

## 2. Results

### 2.1. A Highly Active Bacillus Strain Catabolizes ZEN into ZEM

One-hundred and ninety-four single strains—a culturable microbiome—were obtained from the rhizosphere soil of *P. heterophylla*. They were constructed using a gradient dilution coating method as previously described [36]. The culturable microbiome was used to screen ZEN-degrading strains based on the mixed single-bacterial hierarchical screening method as previously described [36]. The ZEN-degradation rate of the N3-3 consortium reached 46.74%, which was significantly higher than the other eight bacterial consortiums divided from the culturable microbiome (*p* < 0.05). Then, the N3-3 consortium was divided into four sub-consortiums; the N3-3-D sub-consortium had a 61.63% ZEN degradation rate, which was higher than the other three sub-consortiums (*p* < 0.05). The strain D-1 from the N3-3-D sub-consortium with a 73.92% ZEN degradation rate was significantly higher than D-2, D-3, D-4, and D-5 strains (*p* < 0.05), displayed stable ZEN-degradation activity and was selected for subsequent study (Figure 1A,H).

The identity of the gram-positive D-1 strain was determined by sequencing its V3V4 rDNA gene (GenBank ID in NCBI: OR487489) and performing BLAST searches against V3V4 rDNA sequences retrieved from NCBI. The D-1 rDNA sequence had the highest identity (99.30%) of the rDNA of *Bacillus amyliquefaciens* (GenBank ID in NCBI: MH997538.1) (Figure 1B). Thus, we conclude that strain D-1 is a member of the genus *Bacillus*.

It has been reported that temperature, treatment times, bacterial initial concentration, and pH can influence the degradation of deoxynivalenol (DON) with *Lactobacillus plantarum* [37]. We tested the ZEN degradation rate with D-1 under different bacterial inoculation amounts (volume %), pH values, treatment times, and treatment temperatures. It was found that all of these factors affected the zearalenone removal rate through D-1 (Figure 1).

To evaluate the ZEN degradation activity of the D-1 strain versus time, D-1 was grown in LB containing ZEN (5 μg·mL^−1^), and changes in ZEN concentration were monitored every 12 h using HPLC. The degradation rate of ZEN sharply increased at 12 h after inoculation (hai), and reached a relatively stable and higher value (80%) at 24 hai (Figure 1C).

ZEN degradation activity of the D-1 strain was also estimated versus temperature; D-1 was grown in LB containing ZEN (5 μg·mL^−1^), incubated at 15, 25, 28, 30, 37, and 45 °C, and changes in ZEN concentration were monitored at 24 hai using HPLC. When the treatment temperature was below 25 °C, the ZEN degradation rate sharply increased with increasing the treatment temperature after treatment at pH 7.0 for 24 h, and reached a relatively stable and higher value (90%), increasing from 25 °C to 30 °C. However, the ZEN degradation rate was less than 80% when the treatment temperature was more than 45 °C (Figure 1D).

The initial pH of the culture medium also affected the ZEN degradation rate with D-1 (Figure 1E). Treating 5 μg·mL^−1^ ZEN with a 1% inoculation amount of (OD_600_ = 1) D-1 at pH 6.0 and 7.0 for 24 h resulted in a significantly higher ZEN removal rate (nearly 90%) than that at pH 4.0, 5.0, 8.0, 9.0, and 10.0 (<75%) (*p* < 0.05).

ZEN degradation activity of the D-1 strain was evaluated versus inoculation amount; D-1 with 0.5%, 1%, 2%, 5%, and 10% inoculation amount were grown in LB containing ZEN (5 μg·mL^−1^), and changes in ZEN concentration were monitored at 24 h using HPLC. When the inoculation amount is below 2%, the degradation rate of ZEN significantly increases dependent on the manner of the inoculation. However, the degradation rate of ZEN reached a higher value (90%) at 2% (Figure 1F).

To evaluate the ability of D-1 to degrade DON in agricultural products, coix semen contaminated with ZEN was ground, incubated with D-1, and monitored using HPLC for ZEN degradation. Coix semen samples contained 90 μg·g^−1^ of ZEN before treatment, and after incubation with D-1 for 24 h, the ZEN degradation reached 96.7% (Figure 1G). In contrast, levels of ZEN in control samples that were incubated in coix semen without D-1 were not significantly reduced after 24 h. These results demonstrate that strain D-1 can almost eliminate DON from coix semen.

To detect the products of ZEN degradation, metabolites from LB cultures containing 5 μg·mL^−1^ ZEN were incubated with (D-1+ZEN group) or without D-1 (ZEN group), and LB cultures without ZEN were incubated with D-1 (D-1 group) and were extracted and analyzed using HPLC. In the D-1+ZEN group, there was one new compound with the retention time of 4.35 min (ZEN-metabolite designated ZEM). The peak area of ZEN with the retention time of 5.15 min in the D-1+ZEN group is significantly lower than that in the ZEN group (Figure 1H). These results indicate that the strain D-1 catabolizes ZEN into a new compound ZEM.

### 2.2. Analysis of the New ZEN-Derived Bacterial Metabolite Using UPLC-Q/TOF MS/MS

Compound ZEM was purified using semi-preparative HPLC, and, along with ZEN, was subjected to analysis with high-performance liquid chromatography/time of flight mass spectrometry (HPLC-Q/TOF MS/MS) to determine its identities. ZEN (3.62 min) and its catabolite (3.25 min) had a distinct retention time (Figure 2A), confirming that they are structurally distinct. The molecular mass for each derivatized compound was determined from the primary mass ion: ZEN-1H^+^ is 316.8507 Da at negative mode (the molecular mass of ZEN is 318.1467 Da), and compound ZEM-1H^+^ is 500.6810 Da at negative mode (the molecular mass of ZEM is 502.3043 Da, the difference of ZEM derivatized from ZEN was nearly 184 Daltons to ZEN) (Figure 2B).

The new compound ZEM was further characterized by a secondary mass ion (Figure 2C). Mass spectra showed fragments at m/z 114.7, 130.0, and 290.8 (Figure 2C), which matched with the m/z fragment of proline (the molecular mass is 115 Da) [38], isoleucine (the molecular mass 131 Da) [39], and 1-(3,5-dihydroxyphenyl)-10′-hydroxy-1-undecen-6-one (the molecular mass 292 Da) [27] as previously published, respectively. Previous research found that ZEN is glycosylated using UDP-glucosyltransferase at C2 and C4 hydroxyl to form ZEN-2,4-di-glucosides [28,40,41]. And, the ester group in the lactone ring of ZEN is easily hydrolyzed using esterase or acid-base hydrolysis to open it, and then is decarboxylated using a decarboxylase to produce a kind of 1-(3,5-dihydroxyphenyl)-10′-hydroxy-1-undecen-6-one which has no estrogenic toxicity [27]. Thus, we speculate that the D-1 strain can transform ZEN to 1-(3-prolinel,5-isoleucinelphenyl)-10′-hydroxy-1-undecen-6-one or 1-(3-isoleucinel,5-prolinelphenyl)-10′-hydroxy-1-undecen-6-one (designated ZEM) with hydrolyzation and decarboxylation at the ester group in the lactone ring and amino acid esterification at hydroxyl of C2 and C4 of ZEN (Figure 2D).

### 2.3. ZEM Has Decreased the Leakage of LDH, the Content of Intracellular MDA, and the Activity of Intracellular SOC of ZEM on TM4 and HepG2 Cell

To confirm the half-inhibitory concentration (IC_50_) of ZEN, cell viability was detected in TM4 and HepG2 cells treated with ZEN and ZEM at final concentrations of 0, 1, 2, 5, 10, 20, 50, and 100 μM using the MTT method. As the ZEN concentration increases, the activity of TM4 and HepG2 cells significantly inclines, and especially sharply decreases at 20 μM (Figure 3A and Appendix A), while, with an increase in ZEM concentration, the activity of TM4 and HepG2 cells is not significantly different except at 100 μM in the HepG2 cell (Figure 3A and Appendix A). The IC_50_ of ZEN to TM4 and HepG2 were calculated as 20 μM. So, we selected 5, 20, and 50 μM concentrations of ZEN and ZEM to further cytotoxicity studies.

To estimate the effect of ZEN and ZEM on the cell growth and morphology (the complement of the cell membrane and nucleus), the cell-fluorescence-staining assay was conducted on TM4 and HepG2 cells treated with ZEN and ZEM at final concentrations 5, 20, and 50 μM. In the DMSO group, TM4 and HepG2 cells grew soundly with the complete cell membrane (red) and nucleus (blue). Compared with the DMSO group, the ZEN group showed a significant decrease in cell count in both TM4 and HepG2 cells with an increase in ZEN concentration. This result shows that ZEN can induce membrane rupture and nuclear shrinkage, while there was no significant difference in cell number and morphology between the ZEM and DMSO groups (Figure 3B and Appendix A). These results show that ZEM has significantly reduced the impact on growth, and the damage to cell membrane and nucleus DNA on both TM4 and HepG2 cells.

Furthermore, the analysis of LDH leakage was performed by detecting the activity of LDA in the cell culture supernatant. The LDH activity was significantly increased in the ZEN group in both TM4 and HepG2 cells as ZEN in a dose-dependent manner. While there was no significant change in the LDH activity in the ZEM group, and the LDH activity in the ZEM group was significantly lower than that in the ZEN group at 5, 20, and 50 μM in HepG2 cells and at 50 μM in TM4 cells (Figure 3C and Appendix A). LDH, a stable cytoplasmic enzyme, was usually used as a biomarker for the damage of the cell membrane [42]. This result further indicates that ZEM has significantly reduced the damage to the membrane of TM4 and HepG2 cells.

In addition, the content of intracellular MDA and the activity of intracellular SOD were detected using an enzyme-linked immunosorbent assay (ELISA). The intracellular MDA content was significantly increased in the ZEN group in both TM4 and HepG2 cells as ZEN in a dose-dependent manner. Meanwhile, there was no significant change in the intracellular MDA content in the ZEM group, and the intracellular MDA content in the ZEM group was significantly lower than that in the ZEN group at 20 and 50 μM in the HepG2 cell and at 50 μM in the TM4 cell (Figure 3D and Appendix A). The activity of intracellular SOD was significantly increased in the ZEN group in both TM4 and HepG2 cells as ZEN in a dose-dependent manner. Furthermore, there was no significant change in the activity of intracellular SOD in the ZEM group, and the activity of intracellular SOD in the ZEM group was significantly lower than that in the ZEN group at 5, 20 and 50 μM in both the HepG2 and TM4 cell (Figure 3E and Appendix A). MDA and SOD used as a biomarker are produced by the cell to resist oxidative stress [43,44,45]. These results show that ZEM can significantly reduce the damage to nucleus DNA by reducing the oxidative stress response.

### 2.4. Decreased Apoptosis of ZEM on TM4 and HepG2 Cell

Cell apoptosis was performed using an Annexin V-FITC/PI double-staining method via flow cytometry. Q1-UR, Q1-UL, Q1-LR, and Q1-LL represent late apoptotic cells, necrotic cells, early apoptotic cells, and surviving cells, respectively. The apoptosis rate (including late apoptotic cells, necrotic cells, and early apoptotic cells) in the ZEN treatment group was significantly higher than that in the DMSO group in both TM4 and HepG2 cells. The apoptosis rate in the ZEM treatment group is significantly lower than that in the ZEN group, and there was no significant difference between the ZEM and DMSO groups. In particular, the proportion of late apoptotic cells and necrotic cells in the ZEM treatment group was significantly lower than that in the ZEN group (Figure 4A,B and Appendix A).

Further, the intracellular ROS content was detected using the DCFH-DA staining method via flow cytometry. The intracellular ROS content in the ZEN treatment group is significantly higher than that in the DMSO group in both TM4 and HepG2 cells. The intracellular ROS content in the ZEM treatment group is significantly lower than that in the ZEN group, and there was no significant difference between the ZEM and DMSO groups (Figure 4C,D and Appendix A). In addition, the mitochondrial-membrane potential was detected using the JC-1 staining method via flow cytometry. The relative ratio of green fluorescence represented the decrease of mitochondrial-membrane potential in the ZEN treatment group was significantly higher than that in the DMSO group in both TM4 and HepG2 cells. The relative ratio of green fluorescence in the ZEM treatment group was significantly lower than that in the ZEN group, and there was no significant difference between the ZEM and DMSO groups (Figure 4E,F and Appendix A). The decrease or dysfunction of mitochondrial-membrane potential indicated a landmark event in the early stages of cell apoptosis [46]. This result shows that ZEM can decline the cell apoptosis initiation through reducing the production of ROS.

To further estimate whether ZEM reduces cell apoptosis, the expression of apoptosis-related genes (p53, Fas, c-JNK, Caspase-8, Bax, and Bcl-2 in HepG2 cell and CHOP, JNK, Caspase-12, Bax, and Bcl-2 in TM4 cell) was detected using qRT-PCR. The expression of Fax, P53, Caspase-1, Jnk, and Bax genes of HepG2 cell was significantly upregulated with ZEN in a dose-dependent manner in the HepG2 cell; as well as this, the expression of the Bcl-2 gene was significantly downregulated. Meanwhile, ZEM does not induce these gene expressions (Figure 5). Furthermore, the expression of Chop, Jnk, Caspase-12, and Bax genes of TM4 cells was significantly upregulated with ZEN in a dose-dependent manner; as well as this, the expression of the Bcl-2 gene was significantly downregulated, and ZEM does not induce these genes expression (Appendix A). In the P53/JNK pathway of the HepG2 cell, the tumor-suppressor gene p53 manages the cell apoptosis with upregulating genes JNK, Bax, Fas, and Caspase-8 expression and downregulating anti-apoptotic gene Bcl-2 expression. And, in the CHOP/Bax pathway of TM4 cell, it has been reported that the activation of the CHOP gene, the endoplasmic reticulum (ER) stress-induced gene, can trigger cell apoptosis through upregulating genes JNK, Bax, and Caspase-12 expression and downregulating anti-apoptotic gene Bcl-2 expression [47,48,49,50,51,52,53]. These results show that ZEM can decrease cell apoptosis by inhibiting the activation of the ER stress pathway in HepG2 and TM4 cells.

### 2.5. D-1 Strain with No Adverse Impact on Cell

To estimate the adverse impact of the D-1 strain on HepG2 and TM4 cells, the cell viability was performed using the MTT method in the coculture assay. D-1 strains were cocultured with HepG2 and TM4 cells, respectively; after 24 h of coculture, there was no significant difference in cell viability between the D-1 group and the DMSO group in both HepG2 and TM4 cells (Figure 6). This result indicates that the ZEN-detoxifying strain D-1 has no adverse effect on the viability of the HepG2 and TM4 cell, laying a foundation for its application.

## 3. Discussion

In this study, the optimal conditions for D-1 to degrade ZEN were of 1% (OD_600_ = 1) bacterial cells to treat 5 μg·L^−1^ ZEN, at pH 6.0~7.0, and at a temperature of 25 °C for 24 h (Figure 1C). Under these conditions, the highest ZEN-removal rate was 90%, which is much higher than that previously reported, such as of *Lactobacillus plantarum* (<48%) [54], *Bacillus subtilis* (55%) [55,56], *Saccharomyces cerevisiae* (<77%) [57], *Rhodococcus* spp. (<70%) [9], and *Aspergillus niger* (<89%) [58]. *B. amyloliquefaciens* has been reported to be added to infant milk and food as a probiotic additive and would be safe to use in food processing [34]. These results indicate that there is great potential of D-1 for degrading ZEN from food or feed. Certainly, it was still comparable to the reported bacterial strains *B. velezensis* A2 [59] and *B. cereus* BC7 [60]. These results show that D-1 provide an efficient microbial resource for the biological detoxification of ZEN.

The principles of microbial degradation mainly include the biosorption of the mycotoxin onto the walls of microbial cells and biotransformation of the mycotoxin catalyzed with microbial-secreted enzymes [28]. It has been previously reported that *S. cerevisiae*, *Bacillus* spp., and *Lactobacillus* spp. are relatively stable ZEN-absorbing agents [28,34,55,61,62,63], and that the main body responsible for this adsorption is the functional carbohydrates in their cell wall via hydrogen bonding, ionic interactions, or hydrophobic interactions. However, as the adsorption way is an incomplete method for mycotoxin elimination, which is reversible and still exists, and mycotoxin can be rereleased in the intestine tract of humans, this method has no practical application for food or feed detoxification. The biotransformation assay of cell’s different compartments showed that the D-1 strain catalyzed ZEN into a new non-toxic component (ZEM) via an enzyme in intracellular fluid and as it has less biosorption (Appendix A), which is consistent with previous reports that the adsorption capacity of almost all of the *Bacillus* spp. strains is far less important than their degradation effects caused by an enzyme [55]. It demonstrates that the D-1 strain should be paid more attention to when researching and developing their use in degradation-enzyme technology in the future.

It is preciously reported that the degradation components of ZEN mainly include α-ZEL, β-ZEL, α-ZOL, β-ZOL, ZEN-glucosides and -di-glucosides, and ZEN-sulfate, ZOM-1 and 1-(3,5-dihydroxyphenyl)-10′-hydroxy-1-undecen-6-one [4,9,22]. However, D-1 can effectively hydrolyze and decarboxylate the macrolactone of ZEN and amino acid esterification at its C2 and C4 hydroxy to form ZEM, which is completely different from previous degradation products (Figure 2). Hence, it is a new structural metabolite biodegraded from ZEN. These results indicate that D-1 can degrade ZEN into non-mimic toxic metabolites.

It is preciously reported that the toxicity of ZEN is mainly caused by its C2 and C4 hydroxy and macrolactone structures, which resemble estrogen. However, not all ZEN metabolites are effective detoxification products. For instance, degradation products α-ZEL or β-ZEL still have strong estrogenic toxicity, and ZEN-glucosides, ZEN-di-glucosides, and ZEN-sulfate are regarded as mimicked mycotoxin and can be released by the digestive tract. Structural analysis of ZEM revealed that ZEN not only hydrolyzed and decarboxylated the macrolactone structure but also its C2 and C4 hydroxyl groups are esterified with amino acids. Previously, research has reported that the cleavage of the lactone ring of ZEN is considered effective detoxification of ZEN [64]. This result indicates that ZEM can significantly reduce the biological detoxification of ZEN.

In this study, we compared the cytotoxicity between ZEN and its metabolite ZEM on HepG2 and TM4 cells. The activities of LDH, MDA, and SOD can serve as biochemical markers for oxidative damage [59]. It has previously been reported that ZEN can cause a change in MDA content and the SOD activity of these biochemical markers [43,44,45]. In our study, we found that ZEN induced oxidative damage in the cell membrane and nucleus DNA, which is in accordance with previous reports [11]. Additionally, its metabolite ZEM has no damage to the cell membrane and nucleus DNA. These results indicate that the metabolite ZEM cannot induce oxidative damage in the cell membrane and nucleus DNA.

It has previously been reported that the ER stress pathway participates in oxidative damage and apoptosis induced with ZEN [47,48,65,66]. Previous research showed that ER stress induces the protein-folding properties of the endoplasmic reticulum, which leads to the accumulation of unfolded proteins in the ER lumen. This causes the regulatory factor GRP78 to dissociate from PERK, ATF-6, and IRE1, which upregulate downstream transcriptional factors (CHOP, JNK, and Caspase-12) and the pro-apoptotic protein Bax and downregulate antiapoptotic the anti-apoptotic protein Bcl-2 that trigger apoptosis in the kidney cells of mice (TM4 cell) [47,48,49,50,51,52,53]. In the present study, our data demonstrated similar results (Appendix A). Multiple reports have shown that treatment with ZEN also induces ER stress in multiple cell systems [47,48,65,67]. Similarly, we also found that ZEN induces ER stress in HepG2 cells (Figure 5). Conversely, its metabolite ZEM cannot activate the cell apoptosis via inducing an ER stress pathway to resist oxidative stress. In addition, ZEM has almost no inhibition on cell growth. Taken together, these results show that the metabolite ZEM almost lost cytotoxicity to HepG2 and TM4 cells.

It was found that the D-1 strain has no adverse effect on the HepG2 and TM4 cells. Previous research found that *B. amyloliquefaciens* with probiotic potentials is broadly used for infant milk, fermented food, gastrointestinal regulators, and domestic animals feed as additives [31,32,33,34,35]. So, these results demonstrate that D-1 can be added as a probiotic to food/feed for the management of ZEN contamination in the future.

## 4. Conclusions

In the present study, an efficient ZEN-detoxifying microbiota, designated D-1, was identified and characterized as *B. amyloliquefaciens* from the culturable microbiome of *Pseudostellaria heterophylla* rhizosphere soil. The D-1 strain can detoxify ZEN to ZEM, which is a new structural metabolite biodegraded from ZEN, through hydrolyzation and decarboxylation at the ester group in the lactone ring and amino acid esterification at C2 and C4 hydroxy. Notably, ZEM has reduced the impact on the viability, and the damage of cell membrane and nucleus DNA via decreasing oxidative stress, including the leakage of LDH, the content of intracellular MDA, and the activity of intracellular SOC, and can significantly decrease the cell apoptosis via inhibiting the activation of the ER stress pathway in the HepG2 cell and TM4 cell. In addition, it was found that the D-1 strain has no adverse effect on the HepG2 and TM4 cells.

## 5. Materials and Methods

### 5.1. Soil, Chemicals, Media, Kits and Cell

Standard chemicals such as ZEN were purchased from Pribolab (Qingdao, China), of which the purity grade is more than 99%. TM4 cell line and HepG2 cell line were purchased from Pricella (Wuhan, China). Luria–Bertani agar (LB) plates were used to screen microbial cultures for ZEN biotransformation capability, isolate single colonies, and construct a culturable microbiome of rhizosphere soil of *Pseudostellaria heterophylla*. CM-0456 medium with FBS, penicillin, and streptomycin were used to culture the TM4 and HepG2 cell. An annexin V-FITC/PI double-staining cell-apoptosis detection kit (KGA108) was purchased from KeyGEN BioTECH (Nanjing, China). A mitochondrial-membrane-potential assay kit with JC-1 (C2006) and a reactive oxygen species assay kit (S0033) was purchased from Beyotime Biotechnology (Shanghai, China).

### 5.2. Construction of Culturable Microbiome from Rhizosphere Soil of Pseudostellaria heterophylla

The culturable microbiome from the rhizosphere soil of *P. heterophylla* was constructed using a gradient dilution coating method as previously described [36], in which the rhizosphere soil was collected in Shibing County (27°4′21″ N, 108°8′0″ E) in Guizhou Province, China, in May 2019. Rhizosphere soil of *P. heterophylla* was suspended in phosphate-buffered saline (PBS, 50 mL), and after the soil settled, 250 μL of the resulting supernatants diluted 100 times were evenly applied to LB plates. The cultures were incubated at 37 °C for 3 days. The clones were selected and saved in 96-well plates at −80 °C.

### 5.3. Screening of ZEN-Degrading Strain from the Culturable Microbiome

The culturable microbiome (194 strains) was used to screen the ZEN-degrading strain based on the mixed single-bacterial hierarchical screening method as previously described [36]. It was divided into nine bacterial consortiums that were incubated in 1 mL of LB, supplemented with 5 μg ZEN per mL as previously described [17]. The cultures were incubated at 37 °C for 24 h with shaking at 220 rpm. Levels of ZEN in the cultures were monitored using HPLC as described below. The consortiums with decreased ZEN concentrations were selected. Then, single clones with decreased ZEN concentrations were performed using the same method described above from the consortiums with the highest decreased ZEN concentrations. The mono clone with the highest decreased ZEN concentrations was selected.

### 5.4. Taxonomic Characterization of the Isolated Bacterial Strain

The strain D-1 was subjected to gram staining, morphological observation, and 16 S rDNA gene analysis. The V3V4 fragment of the 16 S rDNA gene was amplified by PCR using the primers shown in Appendix A and sequenced to obtain 421 bp of the V3V4 rDNA sequence. Sequences similar to the D-1 V3V4 rDNA were identified using BLAST accessed on 25 June 2022 (National Center for Biotechnology Information, http://www.ncbi.nlm.nih.gov). Neighbor-Joining (NJ); phylogenetic trees of the V3V4 rDNA sequences from 10 bacterial species were constructed using MEGA v7 software (Institute for Genomics and Evolutionary Medicine, Temple University, China). All sequences were retrieved from NCBI (http://www.ncbi.nlm.nih.gov, accessed on 25 June 2022).

### 5.5. The Degradation Activity Analysis of D-1 Strain Versus Time, Temperature, pH, and Inoculation Amount

The concentration of the D-1 strain was adjusted to OD_600_ = 1 using an LB medium. To determine the effects of incubation time, the D-1 strain (1% inoculation amount) was inoculated in LB medium (pH 7) supplemented with 5 μg·mL^−1^ ZEN. The cultures were incubated at 25 °C for 0, 12, 24, 36, 48, 60, and 72 h with shaking at 220 rpm. To determine the effects of temperature, the D-1 strain (OD_600_ = 1, 1% inoculation amount) was inoculated in LB medium (pH 7) contained with 5 μg·mL^−1^ ZEN. The cultures were incubated at 15, 25, 28, 30, 37, and 45 °C for 48 h with shaking at 220 rpm. To determine the effects of pH, the pH of the LB medium was adjusted to 4, 5, 6, 7, 8, 9, and 10. D-1 strain (OD_600_ = 1, 1% inoculation amount) was inoculated in LB medium with different pH values contained with 5 μg·mL^−1^ ZEN. The cultures were incubated at 25 °C for 48 h with shaking at 220 rpm. To determine the effects of inoculation amount, D-1 strain (OD_600_ = 1) according to 0.5%, 1%, 2%, 5%, and 10% inoculation amount was inoculated in LB medium (pH = 4, 5, 6, 7, 8, 9, 10) contained with 5 μg·mL^−1^ ZEN, respectively. The cultures were incubated at 25 °C for 48 h with shaking at 220 rpm. Triplicates were used for each time, temperature, pH, and inoculation amount condition. The ZEN degradation rate was calculated with this formula: reduction in ZEN concentration (%) = (concentration of ZEN original − concentration of ZEN residual)/concentration of ZEN original × 100.

### 5.6. Assay of D-1 ZEN-Degrading Activity in Coix Semen Contaminated with ZEN

Coix semen was ground, passed through a 40-mesh screen, and then autoclaved at 121 °C for 18 min. One gram of autoclaved coix semen sample supplemented with ZEN standards at 90 μg·g^−1^ concentration was mixed with 1 mL LB with or without *Bacillus amyloliquefaciens* D-1 strain (1 × 10^10^ CFU·mL^−1^) in a 2 mL tube, and the mixture culture was incubated at 25 °C for 48 h in an aerobic chamber [68]. The samples were freeze-dried. ZEN was extracted and assayed using HPLC as described below.

### 5.7. Analysis of the ZEN-Degradation of D-1 Is via Enzyme Conversion

The D-1 strain was cultured in a 100 mL flash at 25 °C for 48 h with shaking at 220 rpm. The extracellular fluid and cells of D-1 were collected by centrifuging at 12,000× *g* for 10 min at 4 °C. The cell that was washed with PBS three times was homogenated for 30 min at 4 °C using ice bath sonication methods (frequency 20 kHz; power 150 W; every cycle includes ultrasound 10 s and interval 10 s). Next, intracellular fluid and cell debris were collected from the cell homogenate at 12,000× *g* for 10 min at 4 °C. The extracellular fluid and intracellular fluid were filtered with a 0.22 μm microporous filter membrane (Tianjin Navigator Lab Instrument Co., Ltd., Tianjin, China) to remove bacteria, and were used for biotransformation assay as above. Further, to clarify whether D-1 strain degradation of ZEN is via the enzyme, the intracellular fluid treated with protease K at 58 °C for 30 min and boiled for 30 min was used for biotransformation assay as above.

### 5.8. UPLC- qTOF-MS/MS Analysis

UPLC- qTOF-MS/MS analysis of the purified compounds of ZEM was performed using a UPLC coupled with Xevo G2 Q-TOF MS (Waters Acquity UPLC surveyor autosampler system; Xevo G2 Q-TOF MS ion trap mass spectrometer via an electrospray ion source, Milford, MA, USA) as previously described [69], with some modifications. Source settings used for the ionization of constituents were as follows: nebulizer pressure, 30.00 psi; drying gas, 12.0 L·min^−1^; capillary voltage, 2000 V; gas temperature, 350 °C; fragmentor voltage, 200 V; and skimmer voltage, 60 V. Nitrogen (>99.99%) and He (>99.99%) were used as sheath and damping gases, respectively. A full scan of ions ranging from m/z 100 to 1000 in the negative ion mode and the positive ion mode was carried out. The fragment ions were obtained using a collision energy of 35% for both MS^2^ and MS^3^ experiments. Analyses were conducted at ambient temperature, and the data were operated on the Xcalibur software (version 2021). The LOD and LOQ of this method were 0.026 and 0.053 µg·kg^−1^, respectively.

### 5.9. Cytotoxicity Analysis of ZEN and Degradation Products to TM4 and HepG2 Cell

#### 5.9.1. Cell Culture and Experiment Design

TM4 cells and HepG2 cells were cultivated in CM-0456 media containing FBS, penicillin, and streptomycin, at 37 °C and 5% CO_2_ at constant temperature. After about 90% of cells adhered to the wall, we sucked out the culture medium and washed it three times with PBS. Then, the cells were digested with an equivalent trypsin, and when all of them fell off quickly added an equal volume of CM-0456 media to terminate the digestion reaction. The digested cells were transferred to a centrifuge tube and centrifuged at 1200× *g* for 5 min at 4 °C. After centrifugation, the supernatant was removed and an appropriate volume of CM-0456 media was added to adjust the concentration of TM4 cells and HepG2 cells to 1 × 10^5^ mL^−1^.

We added an appropriate amount of TM4 cells and HepG2 cells to the culture plate and cultured at 37 °C and 5% CO_2_. When the cells adhered to the wall, we added ZEN and ZEM with final concentrations of 5, 20, and 50 µM, respectively, with 24 h further cultivation, and DMSO treatment as the control group. Each treatment had six replicates.

#### 5.9.2. MTT Assay for Detecting Cell Viability

Detection of cell viability was performed as previously described [70]. After 24 h treatment, TM4 cells and HepG2 cells were inoculated with 10 μL MTT (3-(4,5)-dimethylthiahiazo (-z-y1)-3,5-di-phenytetrazoliumromide) solution for 4 h. And then the supernatant was sucked off, and TM4 cells and HepG2 cells were inoculated with 110 μL Formazan solution and shaken at low speed for 10 min. After the crystal was fully dissolved, the OD_490nm_ was measured at 490 nm, and the cell survival rate was calculated as the following formula: cell survival rate = (experimental group OD_490_ value − DMSO OD_490_ value)/(control group OD_490_ value − DMSO OD_490_ value) × 100%.

#### 5.9.3. Fluorescence-Staining Assay

Immunofluorescence assay was performed as previously described [46]. After 24 h treatment, the supernatant culture medium was aspirated and the cells were washed three times with PBS buffer. The cells were fixed with 4% paraformaldehyde and stored in tin foil at 4 °C for 12 h. The fixed solution of polymethanol was aspirated and washed with PBS buffer. Then, cell membrane staining solution was added and gently shaken at 50 rpm in the dark at 37 °C for 20 min. Finally, the cell membrane staining solution was removed and washed three times with PBS, and then an appropriate amount of Hoechse 33342 staining solution was added. After incubation for 5 min in the dark at room temperature, the supernatant solution was aspirated and the cells were washed three times with PBS, and observed under a fluorescence microscope.

#### 5.9.4. Detection of Lactate Dehydrogenase (LDH), Intracellular Malondialdehyde (MDA) Content and Superoxide Dismutase (SOD) Activity

After 24 h treatment, the supernatant was drawn and the LDH was released from cells and was detected according to the instructions of the LDH reagent kit (Solebao, Beijing, China). The cells were collected by centrifuging at 1200× *g* for 5 min at 4 °C and were washed three times with PBS. Then, the cells were fragmented and centrifuged at 12000× *g* for 10 min at 4 °C. After centrifugation, the supernatant was collected. The determination of cell protein concentration referred to the BCA reagent kit (Solebao, Beijing, China), and the intracellular MDA content, and SOD activity was detected and calculated using a microplate reader according to the instructions of the MDA and SOD reagent kits (Solebao, Beijing, China).

#### 5.9.5. Detection of Intracellular Reactive Oxygen Species (ROS) Content, Mitochondrial-Membrane Potential, and Cell Apoptosis using Flow Cytometry

After 24 h treatment at 50 μM, the cells were digested with 0.125% trypsin. When all of them fell off the wall, an equivalent CM-0456 media was added to terminate the digestion reaction. Then, the cells were collected by centrifuging at 1200× *g* for 5 min at 4 °C, the supernatant was carefully removed, and the cells were washed three times with PBS.

Determination of intracellular ROS content was performed using flow cytometry according to the instruction of the reactive oxygen species assay kit (C2006) as previously described [71,72]. The cells were serum-free CM-0456 media with 10 μM DCFH-DA added and incubated at 37 °C for 20 min. Then, the cells were collected by centrifuging at 1200× *g* for 5 min at 4 °C, the supernatant was carefully removed, and the cells were washed three times with PBS. The ROS in cells resuspended with PBS were detected using flow cytometry.

Determination of mitochondrial-membrane potential was performed using flow cytometry according to the instruction of the mitochondrial-membrane potential assay kit with JC-1 dye (C2006) as previously described [72].

Determination of cell apoptosis was performed using flow cytometry according to the instruction of annexin V-FITC/PI double-staining cell-apoptosis detection kit (KGA108) as previously described [73]. The cells were resuspended using 500 μL binding buffer, and then were incubated with 5 μL Annexin V-FITC and 5 μL PI in a dark environment at room temperature for 15 min. Then, cell apoptosis was detected using flow cytometry.

#### 5.9.6. Detection of Apoptosis-Related Genes Using qRT-PCR

After 24 h treatment, the cells were collected by centrifuging at 1200× *g* for 5 min at 4 °C and washed three times with PBS. RNA was isolated from the cells using the Trizol (Invitrogen Life Technologies, Carlsbad, CA, USA) and chloroform extraction method, then purified with the Qiagen RNeasy kit (Takara, Japan). cDNA reverse transcription was performed using a high-capacity cDNA conversion kit (Takara, Shiga, Japan). Quantitative RT-PCR (Bio-Rad CFX 96, Carlsbad, CA, USA) was performed, and the threshold amplification cycle number (Ct) was determined for each reaction in the linear phase of the amplification plot. Each sample was tested in triplicate. The gene’s relative expression was determined using the −ΔΔCt method. Specific mRNAs were amplified using the following primers in Appendix A. The apoptosis-related genes *JNK*, *CHOP*, *Caspase-12*, *Bax*, and *Bcl-2* in TM4 cell, *P53*, *c-JNK*, *Caspase-8*, *Bax*, *Bcl-2*, and *Fas* in HepG2 cell were selected for analysis. β-actin and GAPDH were selected as housekeeping genes for TM4 cells and HepG2 cells, respectively.

### 5.10. Statistical Analysis

All assays were performed in triplicate. The results were analyzed using one way ANOVA for multiple comparisons followed by the Duncan test using Origin software Version 2018 (Origin, Redwood City, CA, USA), with significance levels of 0.05.

## Figures and Tables

**Figure 1 toxins-15-00674-f001:**
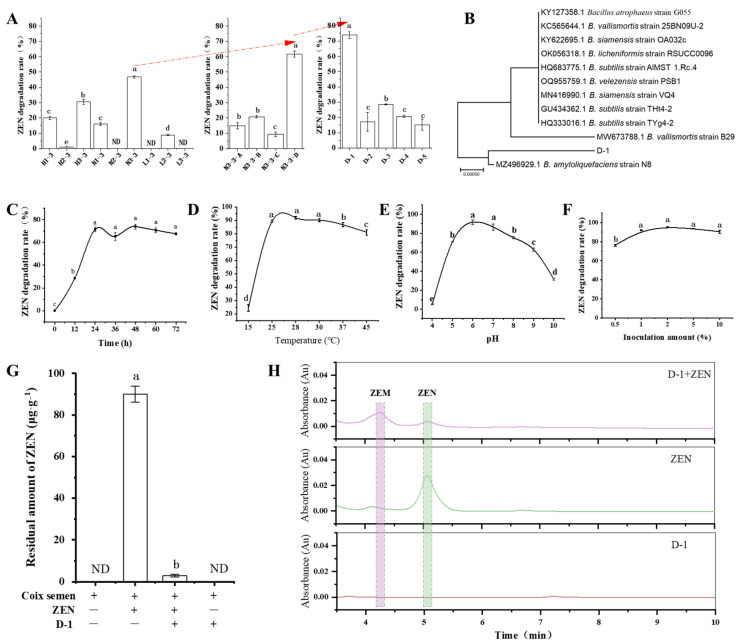
Degradation of zearalenone (ZEN) with *B. amyloliquefaciens* strain D-1. (**A**) The screening of ZEN-degrading strain D-1 based on the mixed single-bacterial hierarchical screening method. (**B**) Phylogenetic tree of *B. amyloliquefaciens* strain D-1 and other homologous strains retrieved from GenBank database, based on V3V4-fragment rDNA sequences. The tree was constructed using neighbor-joining methods, and the scale bar represents the number of substitutions per base position. (**C**–**F**) The ZEN-degradation activity of the D-1 strain was evaluated versus time, temperature, pH, and inoculation amount, respectively. (**G**) The ability of D-1 to degrade ZEN in coix semen; LOQ is 0.053 µg·kg^−1^. (**H**) Determination of ZEN and ZEN metabolites (designated ZEM) using HPLC. ND, not detected. Different lowercase letters indicate significant differences between different data (*p* < 0.05) (n = 3).

**Figure 2 toxins-15-00674-f002:**
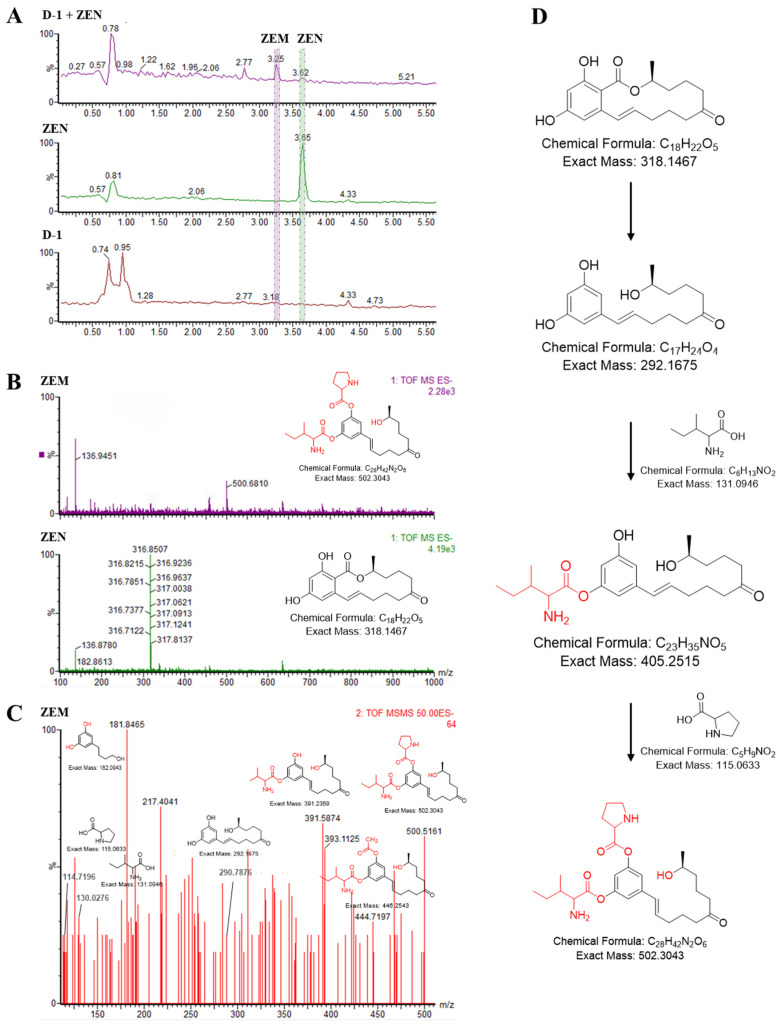
Analysis of the new ZEN-derived bacterial metabolite using UPLC-Q/TOF MS/MS. (**A**) Comparison of total ion peak ZEN metabolite (ZEM) (3.25 min), ZEN (3.62 min), and reagent control group. (**B**) Primary mass spectra of ZEM and ZEN. ZEN is 316.9 Da at negative mode (the molecular mass of ZEN is 318 Da), and compound ZEM is 500.7 Da at negative mode (the difference of ZEM derivatized from ZEN was 184 Daltons to ZEN). (**C**) Secondary mass spectrum of ZEM. (**D**) The proposal degradation diagram of zearalenone with *B. amyloliquefaciens* strain D-1.

**Figure 3 toxins-15-00674-f003:**
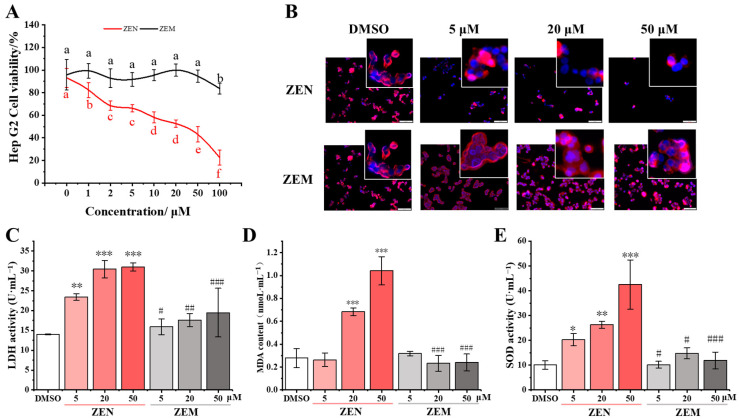
Effects of ZEN and its metabolite ZEM on HepG2 cells. (**A**) Effects of ZEN and ZEM at different concentrations on the activity of HepG2 cells. Letters a, b, c, d, e and f indicate that the ZEM group has a significant difference compared with the ZEN group at *p* < 0.05 levels with the same concentration (n = 6). (**B**) Immunofluorescence staining of HepG2 cells. The cell membrane was stained red with Dil and the nucleus was stained blue with Hoechse 33342. (**C**–**E**) Analysis of the release of LDH, the contents of MDA, and the activity of SOD in HepG2 cells, respectively. *, **, and *** mean that the ZEN group and ZEM group have significant differences compared with the DMSO group at *p* < 0.05, *p* < 0.01, and *p* < 0.005 levels, respectively (n = 6). #, ##, and ### indicate that the ZEM group has a significant difference compared with the ZEN group at *p* < 0.05, *p* < 0.01, and *p* < 0.005 levels with the same concentration, respectively (n = 6).

**Figure 4 toxins-15-00674-f004:**
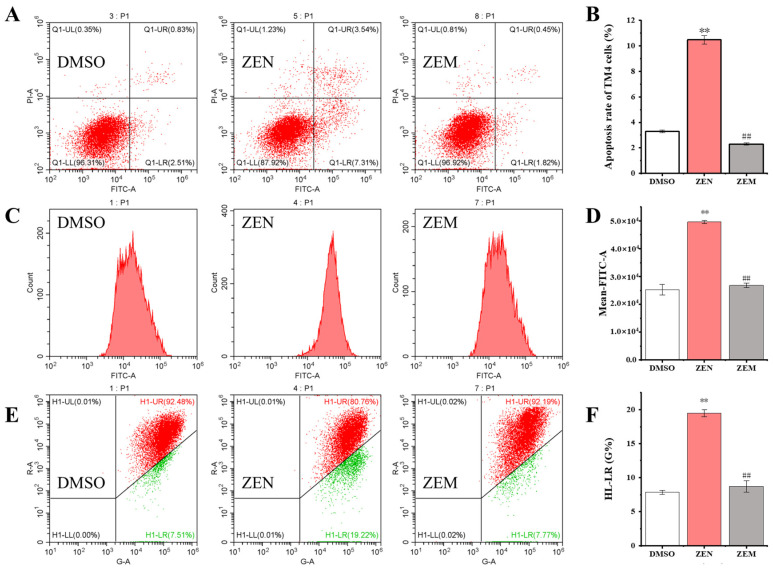
ZEN and its metabolite ZEM impacted on the apoptosis (**A**,**B**), ROS (**C**,**D**), and mitochondrial-membrane potential (**E**,**F**) of HepG2 cells using flow cytometry. The Q1-UL region represents dead cells; the Q1-LR region represents early apoptotic cells; the Q1-LL region represents surviving cells; and the Q1-UR region represents late apoptotic cells. Normal mitochondria fluorescein red, and decreased or lost mitochondrial-membrane potential fluorescein green. ** indicates that the ZEN group and ZEM group have significant difference compared with the DMSO group at *p* < 0.01 level (n = 6). ## indicates that the ZEM group has significant difference compared with the ZEN group at *p* < 0.01 level (n = 6).

**Figure 5 toxins-15-00674-f005:**
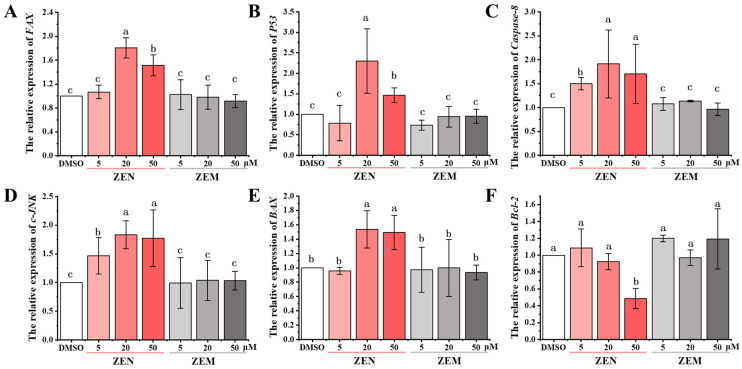
Effects of ZEN and its metabolite ZEM on the expression of apoptosis-related genes in HepG2 cells using qRT-PCR. (**A**), (**B**), (**C**), (**D**), (**E**) and (**F**) indicate genes *FAX*, *P53*, *Caspase 8*, *JNK*, *BAX* and *Bcl-2*, respectively. Different lowercase letters indicated significant differences among different treatment groups (*p* < 0.05) (n = 6).

**Figure 6 toxins-15-00674-f006:**
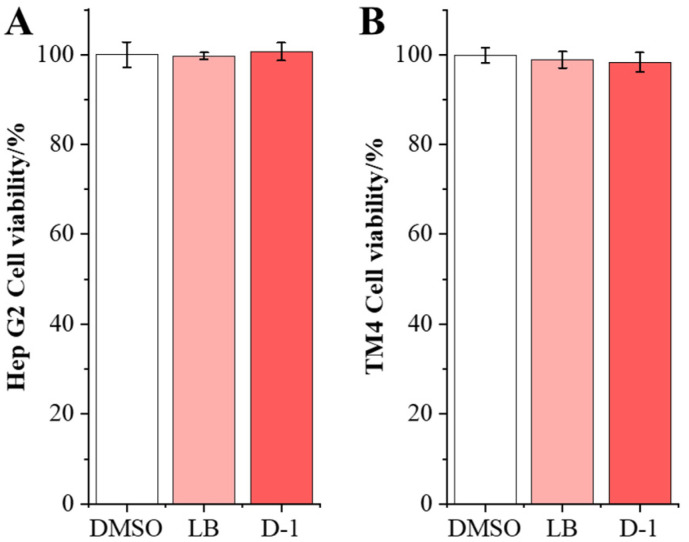
Effects of detoxification strains D-1 on cell activity. (**A**) The viability of HepG2 cells treated with strain D-1 was detected using MTT assay (n = 6). (**B**) The viability of TM4 cells treated with strain D-1 was detected using MTT assay (n = 6).

## Data Availability

Data sharing not applicable.

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
