# Peer review of "A Probiotic Bacillus amyloliquefaciens D-1 Strain Is Responsible for Zearalenone Detoxifying in Coix Semen"

_toxins, 2023, doi:10.3390/toxins15120674_

Round 1

Reviewer 1 Report

Comments and Suggestions for Authors

·        The abstract is quite lengthy and detailed. It should be concise and focused, summarizing the main objectives, findings, and significance of the study in a more succinct manner.

 ·        The introduction lacks context about ZEN, its health risks, and the significance of studying its detoxification. A brief introduction to mycotoxins and their potential harm would be beneficial for readers unfamiliar with the subject.

·        The discussion section could benefit from a more in-depth analysis and interpretation of the results. It should explore the implications of the findings and their relevance to the field of mycotoxin detoxification more thoroughly.

·        The manuscript should discuss how the findings may have practical applications or implications for food and feed safety. How might this research impact real-world practices or future studies in the field?

Comments on the Quality of English Language

Moderate editing of English language required

Author Response

Dear editor and anonymous reviewer,

We thank you very much for helping to improve the manuscript. We have studied all the comments and have incorporated changes to all those we can in this revised manuscript highlight yellow. The followings are detailed point-to-point responses to the comments/suggestions made by you.

Comments and Suggestions for Authors

The abstract is quite lengthy and detailed. It should be concise and focused, summarizing the main objectives, findings, and significance of the study in a more succinct manner.

Response: We have revised the abstract according your suggestion.

The introduction lacks context about ZEN, its health risks, and the significance of studying its detoxification. A brief introduction to mycotoxins and their potential harm would be beneficial for readers unfamiliar with the subject.

Response: That is good suggestion. We have added the introduction context about ZEN in first paragraph in Introduction section.

The discussion section could benefit from a more in-depth analysis and interpretation of the results. It should explore the implications of the findings and their relevance to the field of mycotoxin detoxification more thoroughly.

Response: That is good suggestion. We have added conclusions at each topic in the discussion section in lines 353-354, 368-370, 377-378 according your suggestion.

The manuscript should discuss how the findings may have practical applications or implications for food and feed safety. How might this research impact real-world practices or future studies in the field?

Response: That is good suggestion. We have added a new paragraph to discuss our findings applications in the discussion section in lines 414-418.

Moderate editing of English language required

Response: We have revised throughout MS

We are very sorry for our negligence. We appreciate for Reviewer’s warm work earnestly, and hope that the correction will meet with approval.

Once again, thank you very much for your comments and suggestions.

We are looking forward to your early response.

Kind regards,

Reviewer 2 Report

Comments and Suggestions for Authors

In this paper, the authors present the research results concerning the D1 strain of B. amyloliquefaciens  in detoxifying zearalenone in coix semen. 

The Introduction is Ok and reflects the importance of this research.

At Material and Method, some supplementary information is needed. Where is a place where the sampled rhizosphere soil of Pseudostellaria heterophylla ? What year? Which period of the year?

The rest of the Methodology is well-presented, and no modifications are needed.

Results and Conclusions are ok, and no modifications are needed. 

Comments on the Quality of English Language

English language are ok, only minor speeling modification are needed. 

Author Response

Dear editor and anonymous reviewer,

We thank you very much for helping to improve the manuscript. We have studied all the comments and have incorporated changes to all those we can in this revised manuscript highlight green. The followings are detailed point-to-point responses to the comments/suggestions made by you.

Comments and Suggestions for Authors

In this paper, the authors present the research results concerning the D1 strain of B. amyloliquefaciens in detoxifying zearalenone in coix semen.

Response: Thanks for your comment.

The Introduction is Ok and reflects the importance of this research.

Response: Thanks for your comment.

At Material and Method, some supplementary information is needed. Where is a place where the sampled rhizosphere soil of Pseudostellaria heterophylla? What year? Which period of the year?

Response: We have added the related information you mentioned in lines 447-449.

The rest of the Methodology is well-presented, and no modifications are needed.

Response: Thanks for your comment.

Results and Conclusions are ok, and no modifications are needed.

Response: Thanks for your comment.

English language are ok, only minor speeling modification are needed.

Response: We have revised throughout MS.

We are very sorry for our negligence. We appreciate for Reviewer’s warm work earnestly, and hope that the correction will meet with approval.

Once again, thank you very much for your comments and suggestions.

We are looking forward to your early response.

Kind regards,

Reviewer 3 Report

Comments and Suggestions for Authors

Paragraph between lines 7 and 8 starting with "ZEN with carcinogenity..." should be reworded

Paragraph between lines 80 and 81 starting with "However, the ZEN..." should also be reworded 

Materials and Methods should be moved to the end as required by the journal 

The bibliography should be numbered and inserted in the text as well as required and I also suggest an update of the bibliography with more new titles. 

Comments on the Quality of English Language

Moderate editing of English language required 

Author Response

Dear editor and anonymous reviewer,

We thank you very much for helping to improve the manuscript. We have studied all the comments and have incorporated changes to all those we can in this revised manuscript highlight dark green. The followings are detailed point-to-point responses to the comments/suggestions made by you.

Comments and Suggestions for Authors

Paragraph between lines 7 and 8 starting with "ZEN with carcinogenity..." should be reworded

Response: We have reworded “ZEN with carcinogenicity, genotoxicity, reproductive toxicity, endocrine effects, and immunotoxicity which severely affects food/feed safety and reduces economic losses” into “ZEN severely affects food/feed safety and reduces economic losses owing to its carcinogenicity, genotoxicity, reproductive toxicity, endocrine effects, and immunotoxicity” in lines 43-45.

Paragraph between lines 80 and 81 starting with "However, the ZEN..." should also be reworded

Response: We have reworded it into “However, efficient detoxification microorganisms, transforming ZEN to form non-toxic new metabolites, has rarely been reported” in lines 128-130.

Materials and Methods should be moved to the end as required by the journal

Response: We have revised that.

The bibliography should be numbered and inserted in the text as well as required and I also suggest an update of the bibliography with more new titles.

Response: We have revised that.

Moderate editing of English language required

Response: We have revised throughout MS.

We are very sorry for our negligence. We appreciate for Reviewer’s warm work earnestly, and hope that the correction will meet with approval.

Once again, thank you very much for your comments and suggestions.

We are looking forward to your early response.

Kind regards,

Reviewer 4 Report

Comments and Suggestions for Authors

The readability of this paper is very good and the experimental design is complete and well described. The obtained results are encouraging for possible application in food detoxification. Finally, the references section is useful for the reader.

Some revisions needed:

Line 30: …or postharvest, cereal…

Line 35: …can be severely polluted….

Line 49: Not clear if this value is a limit. Please improve.

Line 60: non-toxic

Lines 74-81: please specify here the focus of this study. Limitations of other strains? Please improve.

Lines 94-96: please specify here the metabolic differences of two compounds.

Line 106: Please add the purity grade.

Please add the uncertainty (where available) for temperature values indicated in the MS.

Line 141: Please specify the supplier of software.

Line 142: Please add this website in references section.

Lines 147, 151, 167 and 157: 25 °C 45 °C (add spacing). Please check and correct throughout the MS.

Line 165: Please explain the choice of such concentration.

Line 152: how adjusted? Please specify.

Line 173: Please add more info about ultrasound methods.

Please use “x g” for centrifugation speed and specify the temperature of centrifugations at lines 171, 240, 246, 258.

Line 175: Please specify the supplier of filter membrane.

Par. 2.8: Please add all available validation parameters (i.e., LOD/LOQ, accuracy, etc.).

Please use past-tense at lines 196-206, 224-235, 450-457.

Line 207: Please specify the meaning of MTT acronym.

Line 271: Please specify what type of ANOVA.

Line 276: Please do not start the sentence using a number.

Line 285: Please specify the p-value.

Line 327: Not all pH < 70%. Please check and modify.

Line 333: There is a decrease for inoculation higher than 2%. Please improve the comment.

Line 357: Please check this value. No link to the Figure.

Please simplify the title of Par. 3.3.

Line 473: …was detected….

Please do not use abbreviations at lines 476 and 479.

Line 502: Figure 6

Figure 1H should be presented alone as Figure 1, Then switching the other parts of figure as figure 2A, 2B…etc. together with other chromatogram examples.

Figure 1G: Please specify the LOQ in the caption.

Figure 6: Please specify the number of repetitions in the caption.

Please check supplementary material which starts with Figure S2.

Please add references at lines 46, 49, 399, 536.

Please arrange the references according to the Journal’s style both in the text and references section.

Comments on the Quality of English Language

Please check and improve readability and correct some typos at lines 5-8, 64-65, 82-83, 196-197, 254, 295, 331, 374, 393, 482, 511.

Author Response

Dear editor and anonymous reviewer,

We thank you very much for helping to improve the manuscript. We have studied all the comments and have incorporated changes to all those we can in this revised manuscript highlight pink. The followings are detailed point-to-point responses to the comments/suggestions made by you.

Comments and Suggestions for Authors

The readability of this paper is very good and the experimental design is complete and well described. The obtained results are encouraging for possible application in food detoxification. Finally, the references section is useful for the reader.

Response: Thank your comments.

Some revisions needed:

Line 30: …or postharvest, cereal…

Response: We have revised it in line 82.

Line 35: …can be severely polluted….

Response: We have revised it in line 86.

Line 49: Not clear if this value is a limit. Please improve.

Response: We have reworded this sentence in lines 98-100.

Line 60: non-toxic

Response: We have revised it in line 110.

Lines 74-81: please specify here the focus of this study. Limitations of other strains? Please improve.

Response: We have improved this part in lines 128-130. However, efficient detoxification microorganisms, transforming ZEN to form non-toxic new metabolites, has rarely been reported.

Lines 94-96: please specify here the metabolic differences of two compounds.

Response: We have specified the metabolic differences of two compounds in lines 146-157. The main difference between these two metabolites is the esterification positions of isoleucine and proline at the C2 and C4 hydroxyl groups.

Line 106: Please add the purity grade.

Response: The purity grade of ZEN is more than 99%, we have added the purity grade of ZEN in lines 433 and 434.

Please add the uncertainty (where available) for temperature values indicated in the MS.

Response: Thank your suggestion, the uncertainty of ZEN standard chemical for temperature values indicated in the MS is not available.

Line 141: Please specify the supplier of software.

Response: We have added the supplier of MEGA 7 software in lines 472 and 473.

Line 142: Please add this website in references section.

Response: We have revised it.

Lines 147, 151, 167 and 157: 25 °C 45 °C (add spacing). Please check and correct throughout the MS.

Response: We have revised that throughout the MS.

Line 165: Please explain the choice of such concentration.

Response: We refer to He et al. 's study published in Toxins to set these concentrations (He et al, 2017) in lines 498-500. I hope our explanation can meet your expectations. 

Line 152: how adjusted? Please specify.

Response: We have revised that in line 477. We diluted the concentration of the D-1 strain to OD600 = 1 using LB medium.

Line 173: Please add more info about ultrasound methods.

Response: We have added Frequency, power, ultrasound cycle of ultrasound methods in lines 504 and 507.

Please use “x g” for centrifugation speed and specify the temperature of centrifugations at lines 171, 240, 246, 258.

Response: We have changed that throughout the MS.

Line 175: Please specify the supplier of filter membrane.

Response: We have added the supplier of filter membrane in lines 510-511.

Par. 2.8: Please add all available validation parameters (i.e., LOD/LOQ, accuracy, etc.).

Response: That's a very good question. We added the LOD/LOQ validation parameters in lines 527-528. We analyzed the LOD/LOQ of UPLC- qTOF-MS/MS method. We found that the LOD and LOQ for ZEN analysis were 0.026 and 0.053 µg·kg-1, respectively. We added the LOD and LOQ for ZEN analysis in lines 255 and 256.

Please use past-tense at lines 196-206, 224-235, 450-457.

Response: We have changed that.

Line 207: Please specify the meaning of MTT acronym.

Response: We have specified the meaning of MTT acronym in lines 546 and 547.

Line 271: Please specify what type of ANOVA.

Response: We have adopted one-way ANOVA for significant analysis. We have specified this info in line 612.

Line 276: Please do not start the sentence using a number.

Response: We have revised that in line 156.

Line 285: Please specify the p-value.

Response: We have specified this information in lines 162, 164, and 166.

Line 327: Not all pH < 70%. Please check and modify.

Response: We have changed < 70% to < 75% in line 196.

Line 333: There is a decrease for inoculation higher than 2%. Please improve the comment.

Response: We have revised this comment in lines 200-202.

Line 357: Please check this value. No link to the Figure.

Response: We have revised the expression to make it more clear in lines 225-228.

Please simplify the title of Par. 3.3.

Response: We have simplified the title of Par. 2.3.

Line 473: …was detected….

Response: We have revised it in line 319.

Please do not use abbreviations at lines 476 and 479.

Response: We have changed that.

Line 502: Figure 6

Response: We have changed it.

Figure 1H should be presented alone as Figure 1, Then switching the other parts of figure as figure 2A, 2B…etc. together with other chromatogram examples.

Response: We have revised it.

Figure 1G: Please specify the LOQ in the caption.

Response: We have specified the LOQ in the caption of Figure 1G.

Figure 6: Please specify the number of repetitions in the caption.

Response: We have specified the number of repetitions of Figure 6.

Please check supplementary material which starts with Figure S2.

Response: We have revised Figure number in supplementary material.

Please add references at lines 46, 49, 399, 536.

Response: We have added the references you mentioned.

Please arrange the references according to the Journal’s style both in the text and references section.

Response: We have rearranged the references according to the Journal’s style both in the text and references section.

Please check and improve readability and correct some typos at lines 5-8, 64-65, 82-83, 196-197, 254, 295, 331, 374, 393, 482, 511.

Response: We have revised the text you mentioned.

We are very sorry for our negligence. We appreciate for Reviewer’s warm work earnestly, and hope that the correction will meet with approval.

Once again, thank you very much for your comments and suggestions.

We are looking forward to your early response.

Kind regards,

Qing-Song Yuan

Round 2

Reviewer 4 Report

Comments and Suggestions for Authors

Minor revision needed.

Please check readability and correct some typos at lines 4-5, 54-46, 71-72, 150, 194, 196-197 (use past-tense), 211, 299, 326 and 560.

A reference is needed at line 350.

Please modify the title of Par. 2.3: ZEM impact on the viability....

Please use past-tense at lines 266-272, 505-511 and 529-530, i.e.:

-  The apoptosis rate (including 266 late apoptotic cells, necrotic cells, and early apoptotic cells) in the ZEN treatment group was 267 significantly higher than that in the DMSO group in both TM4 and HepG2 cells.....

- After centrifugation, the supernatant was removed and an appropriate volume of CM- 506 0456 media was added to adjust the concentration of TM4 cells and HepG2 ....

Lines 293 and 296: ...does not......

Comments on the Quality of English Language

Please check readability and correct some typos at lines 4-5, 54-46, 71-72, 150, 194, 196-197 (use past-tense), 211, 299, 326 and 560.

Author Response

Dear editor and anonymous reviewer,

We thank you very much for helping to improve the manuscript. We have studied all the comments and have incorporated changes to all those we can in this revised manuscript highlight pink. The followings are detailed point-to-point responses to the comments/suggestions made by you.

Comments and Suggestions for Authors

Please check readability and correct some typos at lines 4-5, 54-46, 71-72, 150, 194, 196-197 (use past-tense), 211, 299, 326 and 560.

Response: We have revised those in lines 4-5, 54-56, 71-72, 150-151, 196-197, 211-214, 299-303, 323-329 and 563-565.

A reference is needed at line 350.

Response: We have added the reference in lines 350.

Please modify the title of Par. 2.3: ZEM impact on the viability....

Response: We have revised it according your suggestion.

Please use past-tense at lines 266-272, 505-511 and 529-530, i.e.:

Response: We have revised it in lines, 267, 506-508, and 530-532.

-  The apoptosis rate (including 266 late apoptotic cells, necrotic cells, and early apoptotic cells) in the ZEN treatment group was 267 significantly higher than that in the DMSO group in both TM4 and HepG2 cells.....

Response: We have revised it in line 267.

- After centrifugation, the supernatant was removed and an appropriate volume of CM- 506 0456 media was added to adjust the concentration of TM4 cells and HepG2 ....

Response: We have revised it in lines 506-508.

Lines 293 and 296: ...does not......

Response: We have revised it in lines 293 and 296.

Comments on the Quality of English Language

Please check readability and correct some typos at lines 4-5, 54-46, 71-72, 150, 194, 196-197 (use past-tense), 211, 299, 326 and 560.

Response: We have revised it according your comments.

We are very sorry for our negligence. We appreciate for Reviewer’s warm work earnestly, and hope that the correction will meet with approval.

Once again, thank you very much for your comments and suggestions.

We are looking forward to your early response.
